SciPost Physics

Submission

# Assessing the role of position matrix elements in tight-binding calculations of optical properties

Julen Ibañez-Azpiroz[1*], Fernando de Juan[2,3], Ivo Souza[1,2]

**1** Centro de Física de Materiales, Universidad del País Vasco, 20018 Donostia-San Sebastián, Spain
**2** Ikerbasque Foundation, 48013 Bilbao, Spain
**3** Donostia International Physics Center, 20018 Donostia- San Sebastián, Spain
* julen.ibanez@ehu.es

## Abstract

We appraise the importance of position matrix elements in tight-binding calculations of linear and nonlinear optical properties of acentric materials. A common approximation consists of discarding off-diagonal matrix elements of the position operator $\hat{r}$. Such matrix elements can be naturally incorporated into the tight-binding formalism through the so-called Wannier-interpolation scheme, which we adopt in this work. Using monolayer $BC_2N$ as case study, we find that the shift photocurrent is very sensitive to off-diagonal position matrix elements, which is confirmed in two separate ways: by explicitly evaluating their contribution in *ab initio* calculations, and by means of a $k \cdot p$ model that implicitly assumes the $\hat{r}$-diagonal approximation. Our results indicate that the error incurred by truncating the position matrix is particularly severe for two-band models, where even the linear dielectric function is strongly affected.

# 1  Introduction

The empirical tight-binding (TB) method represents the most natural choice when one aims for an intuitive and transparent representation of the properties of electrons in solids [1]. In the empirical TB method, the basis is defined implicitly through its Hamiltonian matrix elements, without an explicit real-space representation of the basis orbitals [1]. While the Hamiltonian matrix elements are sufficient to obtain the band structure, the calculation of electromagnetic response functions in the dipole approximation requires, in addition, the matrix elements of the position operator [2]. These appear in two places in the calculation: (*i*) in the coupling $\hat{H}' = -e\boldsymbol{E}(t) \cdot \hat{\boldsymbol{r}}$ to the electric field, and (*ii*) in the velocity operator $\hat{\boldsymbol{v}} = (i/h)[H, \hat{\boldsymbol{r}}]$ that is needed to evaluate the induced current.

In empirical TB models, it is customary to make the simplest possible approximation for the position matrix element; namely, to discard all off-diagonal components, keeping the orbital centers as the only spatial information in the model. Formally, this can be expressed as

$$\langle \boldsymbol{0}n|\hat{\boldsymbol{r}}|\boldsymbol{R}m \rangle = \boldsymbol{\tau}_n \delta_{\boldsymbol{R},\boldsymbol{0}} \delta_{nm}, \tag{1}$$

where $|\boldsymbol{R}n\rangle$ denotes a basis Wannier function (WF) in real space with cell index $\boldsymbol{R}$ and intracell orbital index $n$, with $\boldsymbol{\tau}_n$ the center of the $n^{\text{th}}$ WF in the home cell $\boldsymbol{R} = \boldsymbol{0}$. The appeal of the approximation in Eq. (1) is that it allows to express the velocity matrix elements entirely in terms of the Hamiltonian matrix elements. Throughout the text, we will refer to it as the "$\hat{\boldsymbol{r}}$-diagonal approximation" [1].

Despite its appeal, this "minimal spatial embedding" of a TB model is a rather crude approximation. In particular, it discards symmetry-allowed on-site matrix elements such as $\langle s|\hat{x}|p_x \rangle$, as well as all position matrix elements between orbitals in different sites. While the shortcomings of this approximation have been discussed before (e.g., Refs. [2, 4, 5]), its quantitative impact on TB calculations of dielectric and optical properties of solids has not been examined in detail in the literature. To our knowledge, only Ref. [2] has conducted a quantitative analysis along these lines. However, that study was focused exclusively on the *linear* optical conductivity and was based on a simple toy model rather than on a realistic material.

In this work, we assess the contribution of off-diagonal position matrix elements to the linear and nonlinear optical conductivities of crystalline solids using a first-principles method. As a case study, we have selected monolayer $BC_2N$ [6–8]; firstly, because this system breaks

---

[1]In Ref. [3], this was referred to as the "diagonal tight-binding approximation".

inversion symmetry, a necessary condition for observing a quadratic optical response. Secondly, as a two-dimensional (2D) system, the analysis and classification of the different atomic neighbors is simpler than in a bulk system. Thirdly, its simple band-edge structure allows for a complementary study based on a two-band $\boldsymbol{k} \cdot \boldsymbol{p}$ model, which gives further insight into the implications of discarding off-diagonal matrix elements.

Our strategy is as follows. After performing an *ab initio* calculation of the electronic structure, we construct, in a post-processing step, well-localized WFs spanning the bands that participate in the optical transitions under study. We then take those WFs and use them as an orthogonal TB basis to evaluate the band structure and optical conductivities. Since the Wannier orbitals are constructed explicitly, the matrix elements $\langle \boldsymbol{0}n|\hat{\boldsymbol{r}}|\boldsymbol{R}m\rangle$ can be tabulated and included, along with $\langle \boldsymbol{0}n|\hat{H}|\boldsymbol{R}m\rangle$, in the calculation of optical responses; by selectively discarding some or all of the off-diagonal position matrix elements, we are able to assess the impact of different levels of truncation.

The response functions analyzed in this work are the dielectric function and the shift photoconductivity, which are respectively linear and quadratic in the optical electric field. The former is the most fundamental optical coefficient, and the latter was chosen because it is known to probe the spatial structure of the electron wave function encoded in the position matrix element [9]. Indeed, the shift photocurrent arises from a shift in real space of an electron wavepacket during a vertical optical transition in $k$-space; as such, it is expected to be particularly sensitive to the spatial embedding of the TB orbitals. This bears some analogy with the ground state polarization and effective charges in semiconductors, where the TB position matrix elements play a prominent role as well [4]. We anticipate here that the $\hat{\boldsymbol{r}}$-diagonal approximation can introduce considerable errors in the calculation of optical properties, especially in the shift photoconductivity.

The paper is organized as follows. In Sec. 2 we review the basic expressions for the aforementioned optical coefficients, the connection between TB theory and the Wannier interpolation technique we adopt, and the role of the position matrix elements. In Sec. 3 we provide the computational and structural details of our *ab initio* calculations on monolayer $BC_2N$, and in Sec. 4 we present the results of those calculations. Specifically, in Sec. 4.1 we analyze the *ab initio* band structure as well as the optical properties; in Secs. 4.2 and 4.3 we present our main results on the hierarchy of position matrix elements and its effect on optical absorption, respectively; and in Sec. 4.4 we present a description of the band-edge optical response in terms of a two-band $\boldsymbol{k} \cdot \boldsymbol{p}$ model. In Sec. 5, we discuss the results and list our main conclusions. Finally, in Appendix A we use quasi-degenerate perturbation theory to construct two-band $\boldsymbol{k} \cdot \boldsymbol{p}$ models starting from TB (Wannier) Hamiltonians.

## 2    Theoretical framework

Here we describe the Wannier interpolation formalism used to evaluate the length-gauge optical matrix elements in TB theory, and show how the conventional formulation of minimal TB models is recovered in the limit where off-diagonal position matrix elements are discarded.

Let us start by considering the interband contribution to the imaginary (absorptive) part of the dielectric function $\epsilon_{ab}(\omega) = \epsilon'_{ab}(\omega) + i\epsilon''_{ab}(\omega)$ [10],

$$\epsilon''_{ab}(\omega) = \frac{\pi e^2}{\hbar} \int [d\boldsymbol{k}] \sum_{n,m} f_{\boldsymbol{k}nm} K^{ab}_{\boldsymbol{k}nm} \delta(\omega_{\boldsymbol{k}mn} - \omega), \tag{2}$$

where $f_{\boldsymbol{k}nm} = f_{\boldsymbol{k}n} - f_{\boldsymbol{k}m}$ and $\hbar\omega_{\boldsymbol{k}nm} = E_{\boldsymbol{k}m} - E_{\boldsymbol{k}n}$ are differences between occupation factors and band energies, respectively, and the integral is over the first Brillouin zone (BZ), with $[d\boldsymbol{k}] = d^d k/(2\pi)^d$ in $d$ dimensions. In the following, we report values for the relative permittivity $\epsilon''_{ab}/\epsilon_0$. The transition matrix element is given by

$$K^{ab}_{\boldsymbol{k}nm} = r^a_{\boldsymbol{k}nm} r^b_{\boldsymbol{k}mn}, \tag{3}$$

where

$$r^a_{\boldsymbol{k}nm} = (1 - \delta_{nm}) A^a_{\boldsymbol{k}nm} \tag{4}$$

denotes the interband dipole matrix, and

$$A^a_{\boldsymbol{k}nm} = i\langle u_{\boldsymbol{k}n}|\partial_a u_{\boldsymbol{k}m}\rangle \tag{5}$$

the Berry connection matrix, with $|u_{\boldsymbol{k}m}\rangle \equiv e^{-i\boldsymbol{k}\cdot\hat{\boldsymbol{r}}}|\psi_{\boldsymbol{k}m}\rangle$ the cell-periodic part of a Bloch eigenstate $|\psi_{\boldsymbol{k}m}\rangle$ and $\partial_a \equiv \partial/\partial k_a$. Thus, the needed ingredients to evaluate the interband dielectric function are the energy eigenvalues and the matrix elements of the Berry connection between occupied and unoccupied bands.

## 2.1 Wannier interpolation

### 2.1.1 Hamiltonian matrix elements

Assume we have constructed, in a post-processing step, a set of $M$ well-localized WFs per cell $w_j(\boldsymbol{r} - \boldsymbol{R}) = \langle\boldsymbol{r}|\boldsymbol{R}j\rangle$ spanning a set of bands including the initial and final states involved in interband absorption processes up to some desired frequency $\omega$. (With disentangled WFs, the interpolation is faithful only within the so-called "inner" or "frozen" energy window [11].) Starting from these orbitals, we define a set of Blochlike basis states as

$$|u^{(\mathrm{W})}_{\boldsymbol{k}j}\rangle = \sum_{\boldsymbol{R}} e^{-i\boldsymbol{k}\cdot(\hat{\boldsymbol{r}}-\boldsymbol{R}-\boldsymbol{\tau}_j)}|\boldsymbol{R}j\rangle, \tag{6}$$

where the superscript (W) stands for "Wannier gauge" [12].

The matrix elements of the first-principles Hamiltonian $\hat{H}_{\boldsymbol{k}} = e^{-i\boldsymbol{k}\cdot\hat{\boldsymbol{r}}}\hat{H}e^{i\boldsymbol{k}\cdot\hat{\boldsymbol{r}}}$ between the Blochlike states (6) read

$$\begin{aligned} H^{(\mathrm{W})}_{\boldsymbol{k}nm} &= \langle u^{(\mathrm{W})}_{\boldsymbol{k}n}|\hat{H}_{\boldsymbol{k}}|u^{(\mathrm{W})}_{\boldsymbol{k}m}\rangle \\ &= \sum_{\boldsymbol{R}} e^{i\boldsymbol{k}\cdot(\boldsymbol{R}+\boldsymbol{\tau}_m-\boldsymbol{\tau}_n)}\langle\boldsymbol{0}n|\hat{H}|\boldsymbol{R}m\rangle. \end{aligned} \tag{7}$$

Diagonalization of this $M \times M$ matrix yields the Wannier-interpolated energy eigenvalues,

$$\left(U^\dagger_{\boldsymbol{k}} H^{(\mathrm{W})}_{\boldsymbol{k}} U_{\boldsymbol{k}}\right)_{nm} = E_{\boldsymbol{k}n}\delta_{nm}, \tag{8}$$

where $U_{\boldsymbol{k}}$ is a unitary matrix taking from the Wannier gauge to the Hamiltonian gauge. This Slater-Koster type of interpolation, with the WFs acting as an orthogonal tight-binding basis, has been shown to provide a smooth $k$-space interpolation of the *ab initio* band structure.

### 2.1.2 Position matrix elements

The same interpolation strategy can be applied to other $k$-dependent quantities. In particular, the Hamiltonian-gauge Bloch states

$$|u_{\boldsymbol{k}n}\rangle = \sum_{j=1}^{M} |u_{\boldsymbol{k}j}^{(\mathrm{W})}\rangle U_{\boldsymbol{k}jn} \tag{9}$$

interpolate the *ab initio* Bloch eigenstates, allowing to treat wavefunction-derived quantities such as the Berry connection. Inserting Eq. (9) in Eq. (5) yields [12]

$$A_{\boldsymbol{k}nm}^{a} = \mathbb{A}_{\boldsymbol{k}nm}^{a} + \mathrm{a}_{\boldsymbol{k}nm}^{a}, \tag{10a}$$

$$\mathbb{A}_{\boldsymbol{k}nm}^{a} = i\left(U_{\boldsymbol{k}}^{\dagger}\partial_{a}U_{\boldsymbol{k}}\right)_{nm}, \tag{10b}$$

$$\mathrm{a}_{\boldsymbol{k}nm}^{a} = \left(U_{\boldsymbol{k}}^{\dagger}A_{\boldsymbol{k}a}^{(\mathrm{W})}U_{\boldsymbol{k}}\right)_{nm}, \tag{10c}$$

where $A_{\boldsymbol{k}a}^{(\mathrm{W})}$ in Eq. (10c) denotes a Cartesian component of the Berry connection matrix in the Wannier gauge,

$$\begin{aligned} \boldsymbol{A}_{\boldsymbol{k}nm}^{(\mathrm{W})} &= i\langle u_{\boldsymbol{k}n}^{(\mathrm{W})}|\partial_{\boldsymbol{k}}u_{\boldsymbol{k}m}^{(\mathrm{W})}\rangle \\ &= \sum_{\boldsymbol{R}} e^{i\boldsymbol{k}\cdot(\boldsymbol{R}+\boldsymbol{\tau}_m-\boldsymbol{\tau}_n)}\langle\boldsymbol{0}n|\hat{\boldsymbol{r}}-\boldsymbol{\tau}_m|\boldsymbol{R}m\rangle. \end{aligned} \tag{11}$$

Following Ref. [3], we will refer to $\mathbb{A}_{nm}^{a}$ and $\mathrm{a}_{nm}^{a}$ in Eq. (10) as the "internal" and "external" parts, respectively, of the Berry connection matrix in the Hamiltonian gauge. We note that the convention for Bloch sums adopted in Eq. (6), with the Wannier centers included in the phase factors, leads to a rather elegant result in this context: when off-diagonal position matrix elements are discarded the external term vanishes, while the internal term, which can be expressed entirely in terms of Hamiltonian matrix elements, corresponds precisely to the dipole matrix element as usually defined in empirical TB theory [1].

The above suggests a systematic procedure for improving upon the usual empirical TB treatment of optical properties, by adding external terms to the dipole matrix. At first sight, one could expect a natural hierarchy of approximations for the position matrix elements $\langle\boldsymbol{0}n|\hat{\boldsymbol{r}}|\boldsymbol{R}m\rangle$ where the magnitude decays with the distance

$$d_{\boldsymbol{R}nm} = |\boldsymbol{R}+\boldsymbol{\tau}_m-\boldsymbol{\tau}_n|, \tag{12}$$

similar to the hierarchy of on-site and hopping integrals $\langle\boldsymbol{0}n|\hat{H}|\boldsymbol{R}m\rangle$ [13,14]. Intuitively, this would amount to

1. Include diagonal matrix elements only (Wannier centers).

2. Add matrix elements between WFs sitting on the same site.

3. Add nearest-neighbor off-site matrix elements.

4. Add next-nearest-neighbor matrix elements.

5. ...

The reason to expect such hierarchy would be that the overlap between $|\mathbf{0}n\rangle$ and $|\mathbf{R}m\rangle$ generally decays with the distance between the two orbitals; the more localized the WFs, the faster the decay. However, at variance with the Hamiltonian, the position operator acts as a probe of the average location of the WFs involved, which may *increase* with $d_{\mathbf{R}nm}$. Therefore, we anticipate here that the hierarchy of $\langle \mathbf{0}n|\hat{\mathbf{r}}|\mathbf{R}m\rangle$ results from the interplay between these two effects, and is generally different from the hierarchy of $\langle \mathbf{0}n|\hat{H}|\mathbf{R}m\rangle$. In Sec. 4.2, we will analyze in detail the hierarchy of $\langle \mathbf{0}n|\hat{\mathbf{r}}|\mathbf{R}m\rangle$ calculated in monolayer $BC_2N$, and explore how the different levels of truncation impact the calculated optical coefficients.

## 2.2 Shift current and the generalized derivative

We conclude this theory section by considering the shift-current response. This consists of a dc photocurrent density originating form the linear bulk photovoltaic effect [15–17]

$$j^a = 2\sigma^{abb}(0;\omega,-\omega)\mathcal{E}_b(\omega)\mathcal{E}_b(-\omega), \tag{13}$$

with $\mathcal{E}_b(\omega)$ the electric field of light along $b$ at frequency $\omega$. From Ref. [10], the interband (*shift-current*) part of the response reads

$$\sigma^{abb}(0;\omega,-\omega) = \quad -\frac{i\pi e^3}{\hbar^2} \int [d\mathbf{k}] \sum_{n,m} f_{\mathbf{k}nm} I^{abb}_{\mathbf{k}mn} \\ \delta(\omega_{\mathbf{k}mn}-\omega), \tag{14}$$

with the transition matrix element given by

$$I^{abb}_{\mathbf{k}mn} = r^b_{\mathbf{k}mn} r^{b;a}_{\mathbf{k}nm}. \tag{15}$$

Besides the interband dipole matrix, Eq. (15) contains its "generalized derivative"

$$r^{a;b}_{\mathbf{k}nm} = \partial_b r^a_{\mathbf{k}nm} - i\left(A^b_{\mathbf{k}nn} - A^b_{\mathbf{k}mm}\right) r^a_{\mathbf{k}nm} \tag{16}$$

with respect to the crystal momentum $\mathbf{k}$. Like the dipole matrix element itself, its generalized derivative also splits into internal and external terms when expressed in the Wannier representation; for details on this decomposition, we refer the reader to Ref. [3] (see also Ref. [18] for a similar approach).

# 3 Computational and structural details

We have performed density-functional theory calculations of monolayer $BC_2N$ using the `Quantum ESPRESSO` code package [19]. We treated the core-valence interaction using scalar-relativistic projector augmented-wave pseudopotentials taken from the `Quantum ESPRESSO` website. The pseudopotentials were generated with the Perdew-Burke-Ernzerhof exchange-correlation functional [20], and the energy cutoff for the plane-wave basis expansion was set at 70 Ry. The $k$-point mesh used for the self-consistent calculation was $10 \times 10 \times 1$, and for the non-self-consistent step we used a $15 \times 15 \times 1$ mesh. The latter generates the Bloch functions that are used as input for the postprocessing Wannierization procedure. We generated WFs [11,21] via the `Wannier90` code package [22], and computed the dielectric function [Eq. (2)] and the shift-current spectrum [Eq. (14)] using Wannier interpolation. In both cases,

we employed a dense $k$-point interpolation grid of $2000 \times 2000 \times 1$ in order to achieve a well-converged optical spectrum. We used a fixed width of 0.01 eV when broadening the delta functions in Eqs. (2) and (14), which was found to properly handle the van-Hove singularities characteristic of 2D systems.

To model monolayer $BC_2N$, we used a slab geometry with a supercell of length $l = 20$ Å along the out-of-place direction. The 2D crystal structure, illustrated in Fig. 1, is formed by alternating zigzag chains of carbon and boron nitride. The unit cell contains four atoms; the two carbon atoms are inequivalent, and we use symbols $C_N$ and $C_B$ to label the ones with a nitrogen and a boron atom among their nearest neighbors (NNs), respectively. The system has space group $Pmm2$; it is acentric and polar, with the polar axis along $\hat{\mathbf{y}}$, and it contains two mirror operations, $M_x$ and $M_z$. We took the in-plane structural parameters from Ref. [8], with lattice vectors $\mathbf{a}_1 = 2.46\ \hat{\mathbf{x}}$ Å and $\mathbf{a}_2 = 4.32\ \hat{\mathbf{y}}$ Å. The 2D BZ, depicted in Fig. 1, contains four high symmetry points ($\Gamma$, Y, X and S), which are invariant under the two mirror reflections.

For the construction of the WFs we used a one-shot projection method, without any further minimization of the spread functional [21]. In a nutshell, we project a set of $M$ localized orbitals onto the Bloch manifold at every $k$-point, followed by a Löwdin orthonormalization procedure [21, 23]. In this way, we keep as much as possible the atomic character of the trial orbitals, which is desirable for constructing TB models. We note, however, that due to orthogonality requirements the resulting WFs are not pure atomic orbitals. The trial orbitals used throughout the work will be described in Sec. 4.2.

# 4 Results

## 4.1 Electronic and optical properties

The calculated band structure of monolayer $BC_2N$ is displayed in Fig. 2(a), with the direct band gap between valence ($v$) and conduction ($c$) bands plotted in Fig. 2(b). The minimum direct band gap of $E_g \approx 1.6$ eV occurs at the S point, in agreement with previous calculations [6, 8]. In Fig. 3, we show the density of states (DOS) projected onto atomic $p_z$ orbitals on each of the four atoms in the unit cell, compared with the full DOS. The electronic states lying around the Fermi level are composed almost entirely of $p_z$ orbitals, with only a small contribution from other orbitals (mainly $p_x$) at higher energies (not shown). The overall weight of $p_z$ orbitals on different atoms is of the same order in the displayed energy range, with the two inequivalent carbon atoms slightly dominating right at the band edge.

Next we analyze the calculated dielectric function [Eq. (2)] and shift photoconductivity [Eq. (14)]. $BC_2N$ has point group $mm2$, which allows three components of the shift photoconductivity tensor $\sigma^{abb}$ to be nonzero (the full $\sigma^{abc}$ tensor would have five nonzero components): two where the optical field is polarized in plane ($yxx$ and $yyy$), and one where it is polarized out of plane ($yzz$). For the sake of clarity, we will analyze the in-plane components only: $xx$ and $yy$ for the dielectric function, $yxx$ and $yyy$ for the shift photoconductivity. In addition, we will also analyze the behavior of the joint density of states (JDOS) per crystal cell,

$$N(\omega) = \frac{v_c}{\hbar} \int [d\mathbf{k}] \sum_{n,m} f_{nm} \delta(\omega_{mn} - \omega) \tag{17}$$

with $v_c$ the cell volume.

Following Ref. [24], we report a 3D-like response by rescaling the calculated response of the slab of thickness $l$ according to

$$\sigma_{3D}^{abb} = \frac{l}{h} \sigma_{\text{slab}}^{abb}, \tag{18}$$

where $h = \sqrt{\mathbf{a}_1^2 + \mathbf{a}_2^2} = 4.97$ Å is the stacking distance. Throughout the work we report values for $\sigma_{3D}^{abb}$ and omit the 3D subindex, using the notation of Sec. 2. We also rescale the dielectric function by $l/h$.

The calculated JDOS [Fig. 4(a)] exhibits van Hove singularities at $E_g \sim 1.6$ eV and $E_{\Gamma X} \sim 2$ eV, as well as a strong peak at $\sim 2.4$ eV. These features are also present in the two symmetry-allowed in-plane components of the linear dielectric function, as shown in Fig. 4(b). Furthermore, in the band-edge region $\epsilon_{ab}''$ is constrained by dipole selection rules imposed by mirror symmetry $M_x$ [25]. Owing to these selection rules, which are exact right at the band edge and hold to a very good approximation between $E_g$ and $E_{\Gamma X}$, the $yy$ component is negligible over that range while the $xx$ component is sizeable. Above $E_{\Gamma X}$, the $yy$ component becomes sizeable as well.

The shift current spectrum reaches its peak value of $\sigma^{yxx} \sim 60$ $\mu$A/V$^2$ at $\omega \sim 2.4$ eV, which is of the same order as the maximum shift photoconductivity predicted for other 2D materials [26]. $\sigma^{yyy}$, in turn, is significantly smaller than $\sigma^{yxx}$ over the displayed frequency range. In the band-edge region $\sigma^{yyy}$ is virtually zero, as dictated by the same dipole selection rules mentioned earlier [25]. In contrast, $\sigma^{yxx}$ is sizeable and shows a step-like feature, reaching a plateau of $\sim 5$ $\mu$A/V$^2$. Overall, the shapes of $\sigma^{yyy}$ and $\sigma^{yxx}$ as a function of $\omega$ are reminiscent of those of $\epsilon_{yy}''$ and $\epsilon_{xx}''$, respectively.

## 4.2 Hierarchy of position matrix elements

In this section we analyze the magnitudes of the position matrix elements in various WF basis sets. For the WF construction, we have used one-shot projections based on the atomic character of the orbitals near the band-edge [see Fig. 3]. We have considered three different sets of WFs composed of $M = 2$, 4 and 16 bands, using as trial orbitals $p_z$ orbitals centred on the carbon atoms, $p_z$ orbitals centred on every atom, and $s$, $p_x$, $p_y$ and $p_z$ orbitals centred at every atom, respectively. In every case, we have isolated the bands of interest around the Fermi level from the other bands using the "band disentanglement" procedure of Ref. [11]. For $M = 2$, the inner disentanglement energy window [11] included the top of the valence band and bottom of conduction band only, while for the other sets that window was increased to span $\sim 6$ eV. Fig. 5 depicts the WFs obtained with $M = 2$ and $M = 4$.

We will take the set with $M = 4$ WFs as the reference basis for our study, since it captures the dominant orbital character of the band-edge states [see Fig. 5(b)] while covering every atom in the unit cell. In Table 1 we have tabulated the magnitudes of the matrix elements $r_{\boldsymbol{R}nm}^x \equiv \langle \mathbf{0}n|\hat{x}|\boldsymbol{R}m \rangle$ and $r_{\boldsymbol{R}nm}^y \equiv \langle \mathbf{0}n|\hat{y}|\boldsymbol{R}m \rangle$ up to 2$^{\text{nd}}$ NNs, along with the distance between the two orbitals [Eq. (12)].

As anticipated in Sec. 2, the distance between orbitals does not directly correlate with the magnitude of the matrix element. The values for $|r_{\boldsymbol{R}nm}^x|$ in Table 1 show that 1$^{\text{st}}$ NN coefficients are overall one order of magnitude smaller than 2$^{\text{nd}}$ NN coefficients with $n \neq m$, despite the distance between orbitals in the former being roughly half compared to the latter. (2$^{\text{nd}}$ NN coefficients with $n = m$ vanish by symmetry.) [2] Concerning $r_{\boldsymbol{R}nm}^y$, the magnitudes

---

[2]To see this, consider integrals of the type $\int f(x)xf(x - x_0)dx$. By a change of variables, this is seen to

Table 1: Magnitudes of the position matrix elements $r^x_{\boldsymbol{R}nm}$ and $r^y_{\boldsymbol{R}nm}$ up to $2^{nd}$ nearest neighbors (NNs), calculated for the set with 4 WFs. For $2^{nd}$ NNs, terms with $n \neq m$ and $n = m$ have been tabulated separately. For the former, we have indicated in parenthesis the atom that "connects" the two sites; for example, B-$C_N$(N) corresponds to B and $C_N$ atoms that have a common N atom as NN [blue arrows in Fig. 6(a)]. The distance $d_{\boldsymbol{R}nm}$ between orbitals is also given. All quantities are in Angstroms.

| | $d_{\boldsymbol{R}nm}$ | $|r^x_{\boldsymbol{R}nm}|$ | $|r^y_{\boldsymbol{R}nm}|$ |
|---|---|---|---|
| **Wannier centers** | | | |
| B | 0 | 1.23 | 1.18 |
| $C_B$ | 0 | 1.23 | 2.70 |
| $C_N$ | 0 | 0 | 3.39 |
| N | 0 | 0 | 4.77 |
| **$1^{st}$ NN** | | | |
| N-$C_N$ | 1.38 | 0 | 0.0085 |
| $C_B$-$C_N$ | 1.41 | 0.0132 | 0.0026 |
| B-N | 1.44 | 0.0022 | 0.0063 |
| B-$C_B$ | 1.51 | $2 \cdot 10^{-4}$ | 0.0185 |
| **$2^{nd}$ NN $(n \neq m)$** | | | |
| N-$C_B$($C_N$) | 2.41 | 0.0222 | 0.0089 |
| B-$C_N$(N) | 2.45 | 0.0247 | 0.0362 |
| B-$C_N$($C_B$) | 2.52 | 0.0396 | 0.0032 |
| N-$C_B$(B) | 2.56 | 0.0177 | 0.0044 |
| **$2^{nd}$ NN $(n = m)$** | | | |
| B-B | 2.47 | 0 | 0.0486 |
| N-N | 2.47 | 0 | 0.0097 |
| $C_N$-$C_N$ | 2.47 | 0 | 0.0258 |
| $C_B$-$C_B$ | 2.47 | 0 | 0.0376 |

of $1^{st}$ and $2^{nd}$ NN coefficients with $n \neq m$ are roughly the same; in this case, the $2^{nd}$ NN coefficients with $n = m$ are the dominant ones, with a maximum of $|r^y_{\boldsymbol{R}nm}| \simeq 0.05$ Å for boron pairs.

From the differences between $r^x_{\boldsymbol{R}nm}$ and $r^y_{\boldsymbol{R}nm}$, we conclude that the hierarchy of position matrix elements is strongly direction-dependent. In order to better illustrate this notion, in Fig. 6 we have pictorially ordered the coefficients with largest magnitude; for the sake of clarity, we have focused on the boron atom and its largest neighboring coefficients, with panels (a) and (b) depicting those for $r^x_{\boldsymbol{R}nm}$ and $r^y_{\boldsymbol{R}nm}$, respectively. The figures show that the hierarchies are in general different, and that their real-space structure is not obvious *a priori*.

When considering a larger basis set with 16 WFs, the main difference is the inclusion of WFs centred on the same site. However, due to the virtually null weight of orbitals other than $p_z$ in the PDOS [c.f. Fig. 3], those matrix elements are found to be very small. (For a thorough analysis on the effect of on-site position matrix elements on the linear optical conductivity,

---

be equal to $\int f(x - x_0/2)x f(x + x_0/2)dx + x_0/2 \int f(x - x_0/2)f(x + x_0/2)dx$. The first term vanishes because it is odd in $x$, while the second vanishes for orthogonal functions, which is true for our WFs; in this case, $\int f(x)x f(x - x_0)dx = 0$ for $x_0 \neq 0$.

see Ref. [2].) On the opposite limit, the set with 2 WFs centred on the carbon atoms forms a quasi one-dimensional chain along $x$, as depicted in Fig. 5(a). The matrix elements $r^x_{\boldsymbol{R}nm}$ vanish by symmetry; in this simple case, the hierarchy of the surviving matrix elements does correlate with the distance $d_{\boldsymbol{R}nm}$. Note however that this two-band TB model constitutes a rather poor representation of the actual system, as it misses important orbital contributions from boron and nitrogen atoms; as such, it is expected to yield acceptable results at the band edge only.

## 4.3 Effect of position matrix elements on the optical spectra

We now analyze how the truncation of the position matrix affects the calculated optical responses. Before coming to the full optical response functions, it is instructive to look first at the $k$-resolved transition matrix elements. For this purpose, let us define

$$K^{ab}_{\boldsymbol{k}} = \sum_{n,m}^{v,c} K^{ab}_{\boldsymbol{k}mn} \tag{19}$$

$$I^{abb}_{\boldsymbol{k}} = \sum_{n,m}^{v,c} f_{\boldsymbol{k}nm} I^{abb}_{\boldsymbol{k}mn}, \tag{20}$$

where $K^{ab}_{\boldsymbol{k}mn}$ and $I^{abb}_{\boldsymbol{k}mn}$ are the linear and quadratic optical matrix elements defined earlier. Here the summations are over the upper valence and lower conduction bands only, so that the above quantities describe transitions close to the band edge.

In Fig. 7 we show the dominant components $K^{xx}_{\boldsymbol{k}}$ and $I^{yxx}_{\boldsymbol{k}}$ along the Γ–S–Y path that passes through the band edge at S. Results have been obtained using the basis with four WFs per cell, and include a comparison of the full calculated values with those obtained using different levels of approximation on the position matrix element up to 2$^{\text{nd}}$ NN.

We begin by inspecting the crudest $\hat{\boldsymbol{r}}$-diagonal approximation of Eq. (1). For both $K^{xx}_{\boldsymbol{k}}$ and $I^{yxx}_{\boldsymbol{k}}$, it yields values close to the exact results in much of the S–Y line, while visible differences arise around the band edge. At S, the relative error of the $\hat{\boldsymbol{r}}$-diagonal approximation is $\sim 5\%$ for $K^{xx}_{\boldsymbol{k}}$, but reaches $\sim 50\%$ for $I^{yxx}_{\boldsymbol{k}}$. These truncation errors are in line with what was found in Ref. [3] for bulk GaAs. The inclusion of 1$^{\text{st}}$ NN matrix elements barely improves the results, which was to be expected due to their small magnitude [c.f. Table 1]. On the other hand, further adding 2$^{\text{nd}}$ NN coefficients with $n \neq m$ brings $K^{xx}_{\boldsymbol{k}}$ close to convergence, while for $I^{yxx}_{\boldsymbol{k}}$ it reduces the relative error to $\sim 40\%$, still a very large value. Finally, adding 2$^{\text{nd}}$ NN coefficients with $n = m$ does not improve results for $K^{xx}_{\boldsymbol{k}}$, while it strongly reduces the relative error in $I^{yxx}_{\boldsymbol{k}}$ due to the large magnitude of the 2$^{\text{nd}}$ NN $r^y_{\boldsymbol{R}nn}$ coefficients, which enter $I^{yxx}_{\boldsymbol{k}}$ but not $K^{xx}_{\boldsymbol{k}}$.

The effect of truncating the position matrix elements is also visible in the full optical spectra, as shown in Fig. 8 for $\epsilon''_{xx}$ and $\sigma^{yxx}$. The effect is particularly large in the band-edge (grey) region, where the optical responses show a dependence on the different components of the position operator that is in line with the behavior of the transition matrix elements at S in Fig. 7. At higher energies the deviation persists but the relative error is smaller, since $k$-space regions outside the neighborhood of S are less affected by the truncation.

Truncation effects become more pronounced when considering the basis set with only 2 WFs, as shown in Fig. 9. We remind that this set is expected to yield sensible result only at the band-edge S, this is why the "Full" (exact) results only agree there (compare solid

lines in Figs. 7 and 9). In the case of $K_{\mathbf{k}}^{xx}$, the $\hat{\mathbf{r}}$-diagonal approximation underestimates the exact value by only $\simeq 10\%$, but this rather good agreement is accidental; including coefficients up to $3^{\text{rd}}$ NN *increases* the relative error. This is caused by the large spread of the WFs, which makes coefficients beyond $3^{\text{rd}}$ NN necessary. As for the shift photoconductivity, the $\hat{\mathbf{r}}$-diagonal approximation yields a completely wrong value, which even has the wrong sign. Including coefficients up to $3^{\text{rd}}$ NN does not help fix the sign, and the relative error remains very large ($> 300\%$). This suggests that within the two-WF basis, truncating the position matrix becomes completely inadequate for modelling optical properties, especially the shift photoconductivity.

We finish by mentioning that the results obtained with the basis formed by 16 WFs (not shown) are very similar to those obtained with 4 WFs in the frequency range analyzed in Fig. 8. Thus, the cost of going from 4 to 16 WFs per cell is unjustified for the purpose of describing the optical properties up to $\sim 3$ eV, and the 4-WF basis strikes a good balance between accuracy and economy.

### 4.4 Two-band $\mathbf{k} \cdot \mathbf{p}$ model

In this section we construct a two-band $\mathbf{k} \cdot \mathbf{p}$ model from the *ab initio* calculation using quasi-degenerate perturbation theory (see Appendix A), with the aim of reproducing the band-edge electronic and optical properties. This is motivated in part by previous works that have used two-band models to describe the photocurrent of materials with simple band edges, like monolayer GeS in Ref. [24]. Since the $\mathbf{k} \cdot \mathbf{p}$ model is built from Hamiltonian matrix elements only, it implicitly assumes the $\hat{\mathbf{r}}$-diagonal approximation; the present study will therefore serve to complement the analysis carried out in the preceding sections.

A $2 \times 2$ Hamiltonian expanded around a generic $\mathbf{k}$ point can be expressed in terms of the three Pauli matrices $\sigma_i$ and the $2 \times 2$ identity matrix $\mathbb{1}$ as

$$\tilde{H}(\mathbf{k}) = \epsilon_0(\mathbf{k})\mathbb{1} + \sum_i f_i(\mathbf{k})\sigma_i \;\; (i = x, y, z), \tag{21}$$

where $\epsilon_0(\mathbf{k})$ and $f_i(\mathbf{k})$ are real coefficients. The energies of the valence and conduction bands are given by

$$\begin{aligned} E_v(\mathbf{k}) &= \epsilon_0(\mathbf{k}) - \epsilon(\mathbf{k}), \\ E_c(\mathbf{k}) &= \epsilon_0(\mathbf{k}) + \epsilon(\mathbf{k}), \end{aligned} \tag{22}$$

with $\epsilon(\mathbf{k}) = \sqrt{\sum_i f_i(\mathbf{k}) \cdot f_i(\mathbf{k})}$.

Using the Wannier-based quasi-degenerate perturbation theory outlined in Appendix A, we have constructed a two-band model expanded up to second order around the S point. Its band dispersion is given by the dashed lines in Fig. 2. Comparison with the DFT band structure shows a nice quantitative agreement in the neighborhood of the band edge, in line with what was found in a similar Wannier-based study of transition metal dichalcogenides [27].

Next, we analyze the ability of the two-band $\mathbf{k} \cdot \mathbf{p}$ model to predict optical properties. Our focus is on the band edge, where the dielectric function and shift photoconductivity take the form

$$\epsilon_{\mathbf{k}\cdot\mathbf{p}}^{ab}(\omega) = \frac{\pi e^2}{\hbar} K_{\mathbf{k}\cdot\mathbf{p}}^{ab}(\omega) N(\omega), \tag{23}$$

$$\sigma_{\mathbf{k}\cdot\mathbf{p}}^{abb}(\omega) = \frac{\pi e^3}{\hbar^2} I_{\mathbf{k}\cdot\mathbf{p}}^{abb}(\omega) N(\omega), \tag{24}$$

*i.e.*, at each frequency they are given by the product between the transition matrix-elements and the JDOS.

The expressions for the matrix elements read [28]

$$K_{\boldsymbol{k}\cdot\boldsymbol{p}}^{ab}(\omega) = \frac{4}{\omega^2} \sum_i f_{i,a} f_{i,b}, \tag{25}$$

and [24]

$$I_{\boldsymbol{k}\cdot\boldsymbol{p}}^{abb}(\omega) = -\frac{1}{2\omega^3} \sum_{ijm} f_m f_{i,b} f_{j,ab} \varepsilon_{ijm}, \tag{26}$$

with $f_{i,a} \equiv \partial_{k_a} f_i$ and $f_{i,ab} \equiv \partial_{k_a} \partial_{k_b} f_i$ (the derivatives are taken at $\boldsymbol{k} = 0$), and $\varepsilon_{ijm}$ the Levi-Civita symbol. The expressions above have been simplified by noting that $\partial_{k_a}\epsilon = 0$ at $\boldsymbol{k} = 0$, given that the band edge lies at an energy extremum.

Using the extracted coefficients $\epsilon_0(\mathbf{k})$ and $f_i(\mathbf{k})$, we have calculated the dielectric function and the shift photoconductivity within the two-band $\boldsymbol{k} \cdot \boldsymbol{p}$ model, as well as the JDOS, for which only the energies of Eq. (22) are needed. The results have been included as dashed lines in the band-edge region of Fig. 4.

The comparison between the $\boldsymbol{k}\cdot\boldsymbol{p}$ and *ab initio* results for the JDOS in the inset of Fig. 4(a) shows an excellent agreement around the band gap, since the height of the step-like feature at $E_g$ is nicely reproduced. Above the band gap, the *ab initio* JDOS grows monotonically, a feature that cannot be captured by the constant prediction of the model, but the discrepancy is rather small.

Turning next to the dielectric function in Fig. 4(b), the $\boldsymbol{k} \cdot \boldsymbol{p}$ prediction for the dominant $xx$ component matches very well the step-like feature present at the band-edge region in the *ab initio* calculation. Furthermore, thanks to the $1/\omega^2$ factor in Eq. (25) the $\boldsymbol{k} \cdot \boldsymbol{p}$ spectrum reproduces the slight decrease in the dielectric function above $E_g$.

We come now to the shift-current spectrum in Fig. 4(c). In contrast to both the JDOS and the dielectric function, the $\boldsymbol{k} \cdot \boldsymbol{p}$ prediction for the dominant shift-current component at $E_g$ clearly deviates from the *ab initio* result, overshooting it by about a factor of five. This large deviation is hardly surprising; the analysis of previous sections revealed that at the S point the matrix element $I_{\boldsymbol{k}}^{abb}$ is strongly affected by the $\hat{\boldsymbol{r}}$-diagonal approximation, and the $\boldsymbol{k} \cdot \boldsymbol{p}$ model has been expanded precisely around that point.

## 5 Discussion

In this work, we have inspected the role of the TB position-operator matrix element in describing linear and nonlinear optical properties of materials, a subject that has received relatively little attention in the literature. To our knowledge, only Ref. [2] has conducted a similar study; however, that study was based on a simple model, it included solely the linear response, and focused on the role of on-site matrix elements that describe intra-atomic optical transitions. Instead, our analysis has relied on a first-principles-based Wannier framework that is close in spirit to the TB method, we have assessed the impact of off-site matrix elements (inter-atomic transitions), and we have studied a realistic material.

Our first-principles calculations have shown that the hierarchy of position matrix elements $\langle \boldsymbol{0}n|\hat{\boldsymbol{r}}|\boldsymbol{R}m\rangle$ as a function of distance between orbitals is not trivial; it is not necessarily dominated by on-site or nearest-neighbor terms, and it is strongly direction-dependent. This can be

understood qualitatively by noting that while growing distance between the two orbitals decreases their overlap, the weight of the position operator term can nonetheless increase, hence the maximum in $|\langle \mathbf{0}n|\hat{\mathbf{r}}|\mathbf{R}m\rangle|$ comes from the balance between these two effects. We note that this is unlike the hierarchy of the more familiar Hamiltonian matrix elements $\langle \mathbf{0}n|\hat{H}|\mathbf{R}m\rangle$ (onsite and hopping parameters in TB), which very generally decay with growing distance [13,14].

Off-diagonal position matrix elements, which are often discarded in practical calculations (see, *e.g.*, Refs. [29–32] for different materials), turn out to be very important for an accurate description of the optical properties of monolayer $BC_2N$, especially the shift photoconductivity. Indeed, their contribution can be as large as that of Hamiltonian matrix elements, which are commonly assumed to dominate. Position matrix elements are particularly important in minimal two-band models, where they completely govern the band-edge response. This is mainly because WFs in two-band models tend to be more spatially extended than in larger basis sets, hence matrix elements decay more slowly with distance. This suggests that, in general, two-band TB models may not be an adequate choice for an accurate description of the shift current, and presumably of other nonlinear optical properties as well. We note that this holds true even in the case of $\mathbf{k} \cdot \mathbf{p}$ models where the expansion coefficients are extracted from *ab initio* calculations.

Finally, it is worth mentioning that while we have focused on $BC_2N$ to clearly illustrate our point, we have also repeated the analysis on another 2D material, namely GeS, for comparison (this was partially analyzed in a previous paper, Ref. [3]). Our results indicate that the quantitative importance of off-diagonal position matrix elements is overall smaller than in monolayer $BC_2N$. This highlights the strong sensitivity of optical properties to the real-space structure of the WFs, which is highly system-dependent. At any rate, the general conclusions drawn for $BC_2N$ fully hold, providing further evidence of the prominent role of off-diagonal position matrix elements in the TB description of nonlinear optical properties.

*Acknowledgements* – We thank David Vanderbilt and Michele Modugno for discussions, and Stepan Tsirkin for sharing the `irrep` computer code that determines the mirror eigenvalues of the Bloch states [33] as well as for a previous collaboration on related work. This work was supported by Grant No. FIS2016-77188-P from the Spanish Ministerio de Economía y Competitividad. This project has also received funding from the European Union's Horizon 2020 research and innovation programme under the Marie Sklodowska-Curie grant agreement No 839237.

# A    Construction of the $\mathbf{k} \cdot \mathbf{p}$ model

## A.1    Quasi-degenerate (Löwdin) perturbation theory

Consider a Hamiltonian

$$H = H^0 + H' \tag{27}$$

where the eigenvalues $E_n$ and eigenfunctions $|n\rangle$ of $H^0$ are known, and $H'$ is a perturbation. In a nutshell, quasi-degenerate perturbation theory assumes that the set of eigenfunctions of $H^0$ can be divided into subsets A and B that are weakly coupled by $H'$, and that we are only interested in subset A, the "active" subspace. This theory asserts that a transformed Hamiltonian $\tilde{H}$ exists within subspace A such that

$$\tilde{H} = \tilde{H}^0 + \tilde{H}^1 + \tilde{H}^2 + \cdots , \tag{28}$$

where $\tilde{H}^j$ contain matrix elements of $H'$ to the $j$th power. According to Ref [34], the first three terms are

$$\tilde{H}^0_{mm'} = H^0_{mm'}, \tag{29}$$

$$\tilde{H}^1_{mm'} = H'_{mm'}, \tag{30}$$

$$\tilde{H}^2_{mm'} = \frac{1}{2}\sum_l H'_{ml}H'_{lm'}\left(\frac{1}{E_m - E_l} + \frac{1}{E_{m'} - E_l}\right), \tag{31}$$

where $m, m' \in \mathrm{A}$ and $l \in \mathrm{B}$. The approximation $\tilde{H} \approx \tilde{H}^0 + \tilde{H}^1$ amounts to truncating $H$ to the A subspace. By further including $\tilde{H}^2$, the coupling to the B subspace is incorporated approximately, renormalizing the elements of the truncated matrix.

### A.2    Application to Wannier-based $\boldsymbol{k} \cdot \boldsymbol{p}$ perturbation theory

Let us adopt the notation of Ref. [12]. We shift the origin of $k$ space to the point where the band edge (or some other band extremum of interest) is located, and Taylor expand the Wannier-gauge Hamiltonian around that point,

$$H^{(\mathrm{W})}(\mathbf{k}) = H^{(\mathrm{W})}(0) + \sum_a H^{(\mathrm{W})}_a(0)k_a + \frac{1}{2}\sum_{ab} H^{(\mathrm{W})}_{ab}(0)k_a k_b + \mathcal{O}(k^3), \tag{32}$$

where

$$H^{(\mathrm{W})}_a(0) = \left.\frac{\partial H^{(\mathrm{W})}(\mathbf{k})}{\partial k_a}\right|_{\mathbf{k}=0}, \tag{33}$$

$$H^{(\mathrm{W})}_{ab}(0) = \left.\frac{\partial^2 H^{(\mathrm{W})}(\mathbf{k})}{\partial k_a \partial k_b}\right|_{\mathbf{k}=0}. \tag{34}$$

We now apply to $H^{(\mathrm{W})}(\mathbf{k})$ a similarity transformation $U(0)$ that diagonalizes $H^{(\mathrm{W})}(0)$, and call the transformed Hamiltonian $H(\mathbf{k})$,

$$H(\mathbf{k}) = \overline{H} + \sum_a \overline{H}_a k_a + \frac{1}{2}\sum_{ab} \overline{H}_{ab}k_a k_b + \mathcal{O}(k^3), \tag{35}$$

where we have introduced the notation

$$\overline{\mathcal{O}} = U^\dagger(0)\mathcal{O}^{(\mathrm{W})}(0)U(0) \tag{36}$$

and applied it to $\mathcal{O} = H, H_a, H_{ab}$. We can now use the machinery of Sec. A.1 by choosing the diagonal matrix $\overline{H}$ as our $H^0$, and the remaining (nondiagonal) terms in Eq. (35) as $H'$. Collecting terms in Eq. (28) up to second order in $\mathbf{k}$ we get

$$\tilde{H}_{mm'}(\mathbf{k}) = \overline{H}_{mm'} + \sum_a \left(\overline{H}_a\right)_{mm'} k_a + \frac{1}{2}\sum_{a,b}\left[\left(\overline{H}_{ab}\right)_{mm'} + (T_{ab})_{mm'}\right]k_a k_b + \mathcal{O}(k^3), \tag{37}$$

where $m, m' \in \mathrm{A}$ and we have defined the virtual-transition matrix

$$(T_{ab})_{mm'} = \sum_{l \in B}\left(\overline{H}_a\right)_{ml}\left(\overline{H}_b\right)_{lm'} \times \left(\frac{1}{E_m - E_l} + \frac{1}{E_{m'} - E_l}\right) = (T_{ba})^*_{m'm}. \tag{38}$$

## A.3 Two-band case

Let us now specialize to the case where the active subspace comprises only two bands. Then $\tilde{H}(\mathbf{k})$ is a $2 \times 2$ Hermitean matrix, and can be expanded as in Eq. (21). Using $\text{Tr}(\sigma_i) = 0$ and $\text{Tr}(\sigma_i \sigma_j) = 2\delta_{ij}$, we find that the expansion coefficients are given by

$$\epsilon_0(\mathbf{k}) = \frac{1}{2}\text{Tr}\left[\tilde{H}(\mathbf{k})\right], \tag{39}$$

$$f_i(\mathbf{k}) = \frac{1}{2}\text{Tr}\left[\tilde{H}(\mathbf{k}) \cdot \sigma_i\right]. \tag{40}$$

The above quantities and their derivatives can be used to compute the shift-current and dielectric matrix-elements of a generic two-band model; these were expressed in Eqs. (26) and (25) of the main text and in terms of $f_i$, $f_{i,a} \equiv \partial_{k_a} f_i$, and $f_{i,ab} \equiv \partial_{k_a} \partial_{k_b} f_i$. To evaluate these coefficients we need

$$f_i(0) = \frac{1}{2}\text{Tr}\left[\overline{H} \cdot \sigma_i\right], \tag{41}$$

$$f_{i,a}(0) = \frac{1}{2}\text{Tr}_A\left[\overline{H}_a \cdot \sigma_i\right], \tag{42}$$

$$f_{i,ab}(0) = \frac{1}{2}\text{Tr}_A\left[\frac{1}{2}\left(\overline{H}_{ab} + \overline{H}_{ba} + T_{ab} + T_{ba}\right) \cdot \sigma_i\right], \tag{43}$$

where $\text{Tr}_A$ denotes a trace using the $2 \times 2$ blocks of $\overline{H}_{ab}$ and $T_{ab}$ in the A sector, and the Eqs. (42) and (43) are obtained inserting Eq. (37) in Eq. (40) and taking derivatives. Using $\overline{H}_{ab} = \overline{H}_{ba}$ together with

$$\text{Tr}_A\left[T_{ba} \cdot \sigma_i\right] = \text{Tr}_A\left[\sigma_i \cdot T_{ba}\right] = $$
$$\text{Tr}_A\left[\left(T_{ab} \cdot \sigma_i\right)^\dagger\right] = \left(\text{Tr}_A\left[\sigma_i \cdot T_{ab}\right]\right)^*, \tag{44}$$

Eq. (43) can be simplified to

$$f_{i,ab}(0) = \frac{1}{2}\text{Re}\,\text{Tr}_A\left[\frac{1}{2}\left(\overline{H}_{ab} T_{ab}\right) \cdot \sigma_i\right]. \tag{45}$$

The real quantities $f_i(0)$, $f_{i,a}(0)$ and $f_{i,ab}(0)$ given by Eqs. (41), (42) and (45) are sufficient to evaluate the transition matrix elements of Eqs. (25) and (26) in the main text.

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

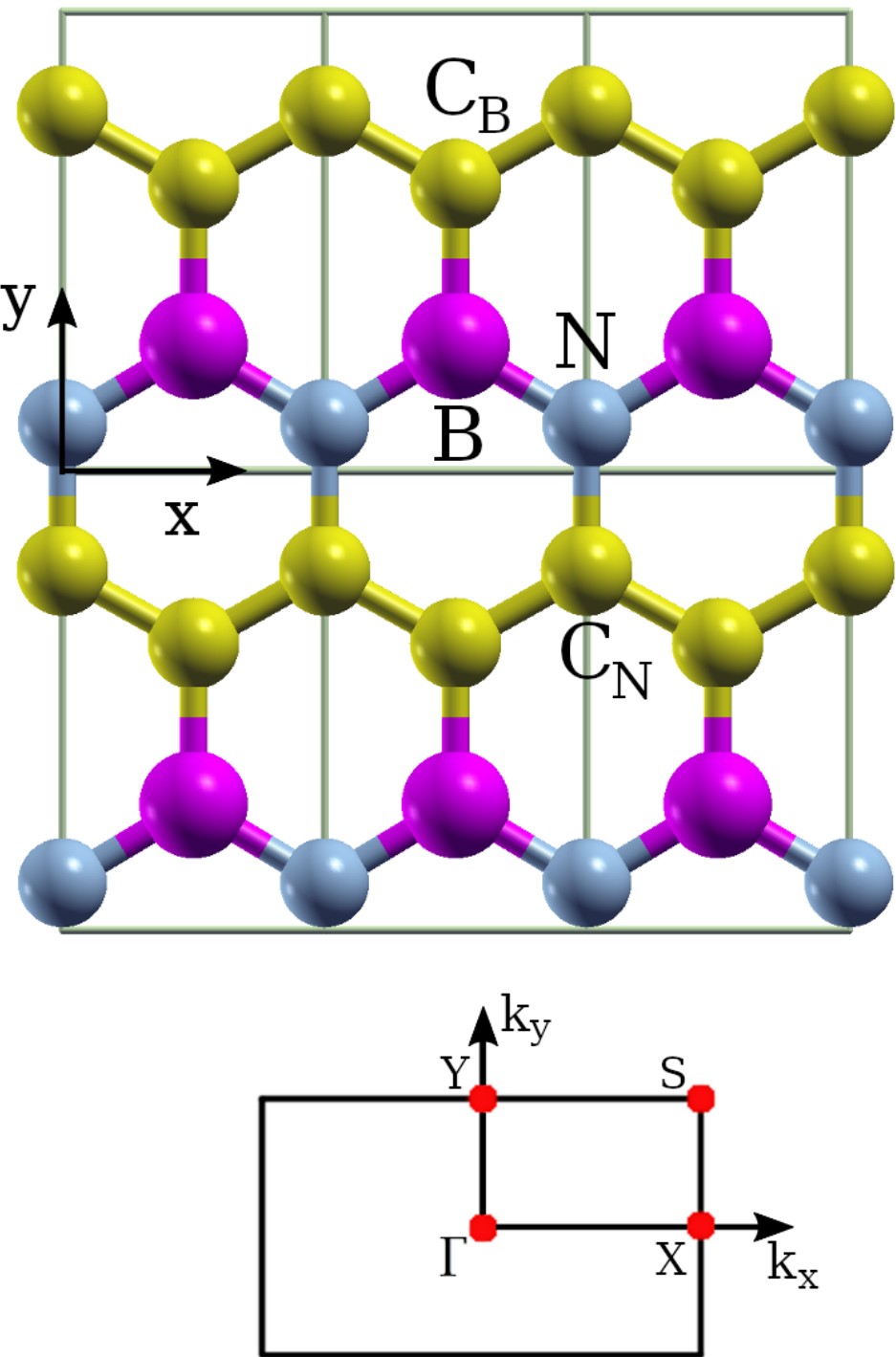

Figure 1: Crystal structure and Brillouin zone of monolayer $BC_2N$. There are two formula units per cell, with carbon, boron and nitrogen atoms represented by yellow, blue and magenta balls respectively. The two carbon atoms in the unit cell are inequivalent; we use the symbols $C_N$ and $C_B$ to label the ones with a nitrogen and boron atoms among their nearest neighbors, respectively. The solid lines delimit the cells, and the vertical lines have been chosen to coincide with one family of mirror planes $M_x$.

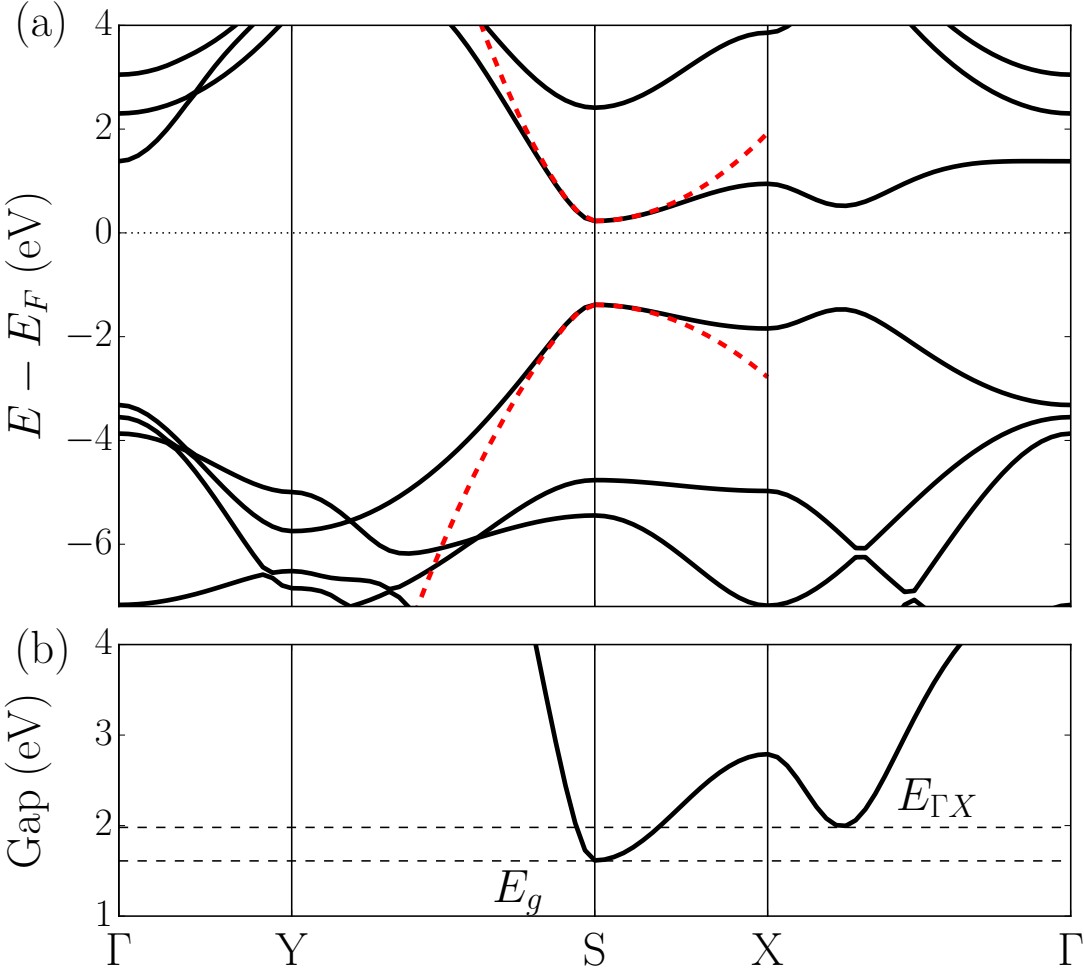

Figure 2: (a) Solid (black) lines denote the *ab initio* band dispersion across the 2D BZ for BC$_2$N. Red dashed lines denote the dispersion obtained from a two-band $\boldsymbol{k}\cdot\boldsymbol{p}$ model expanded around the band edge, as described in Sec. 4.4. Energies are measured from the Fermi level. (b) $\boldsymbol{k}$-resolved value of the direct energy gap between conduction and valence bands. $E_g$ denotes the absolute minimum value, and $E_{\Gamma X}$ marks the minimum energy gap along the $\Gamma X$ line.

 

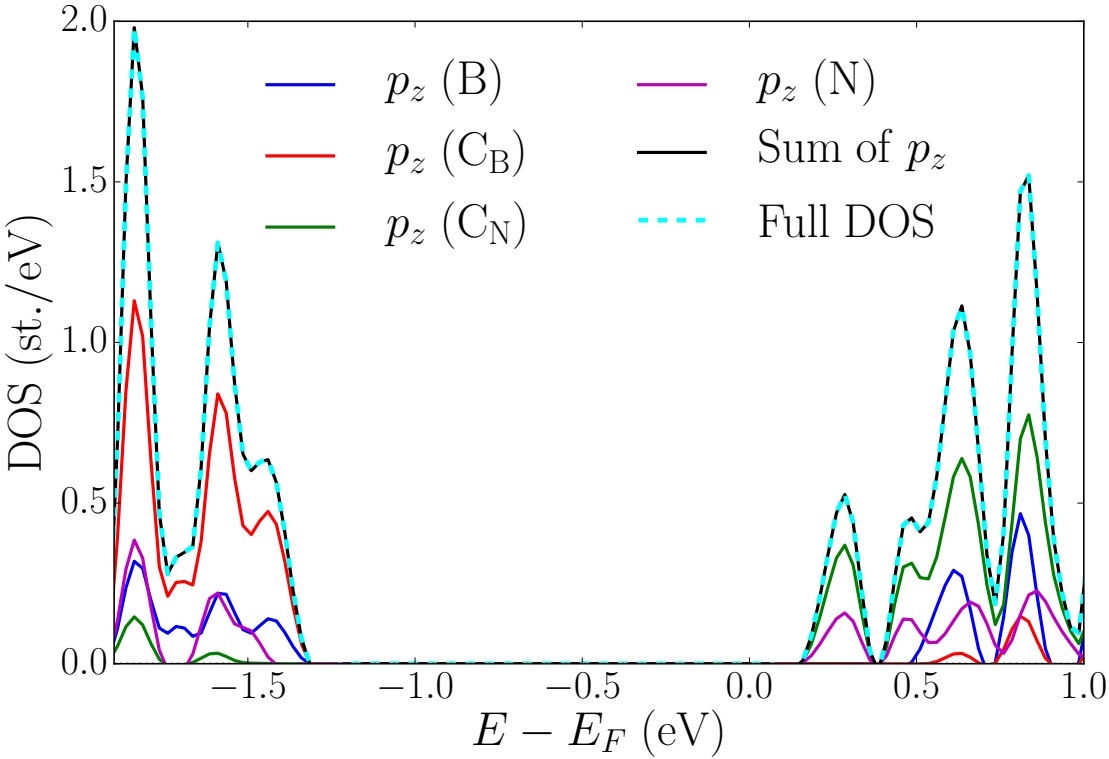

Figure 3: Density of states (DOS) projected onto $p_z$ orbitals on each atom in the unit cell, compared with the full DOS.

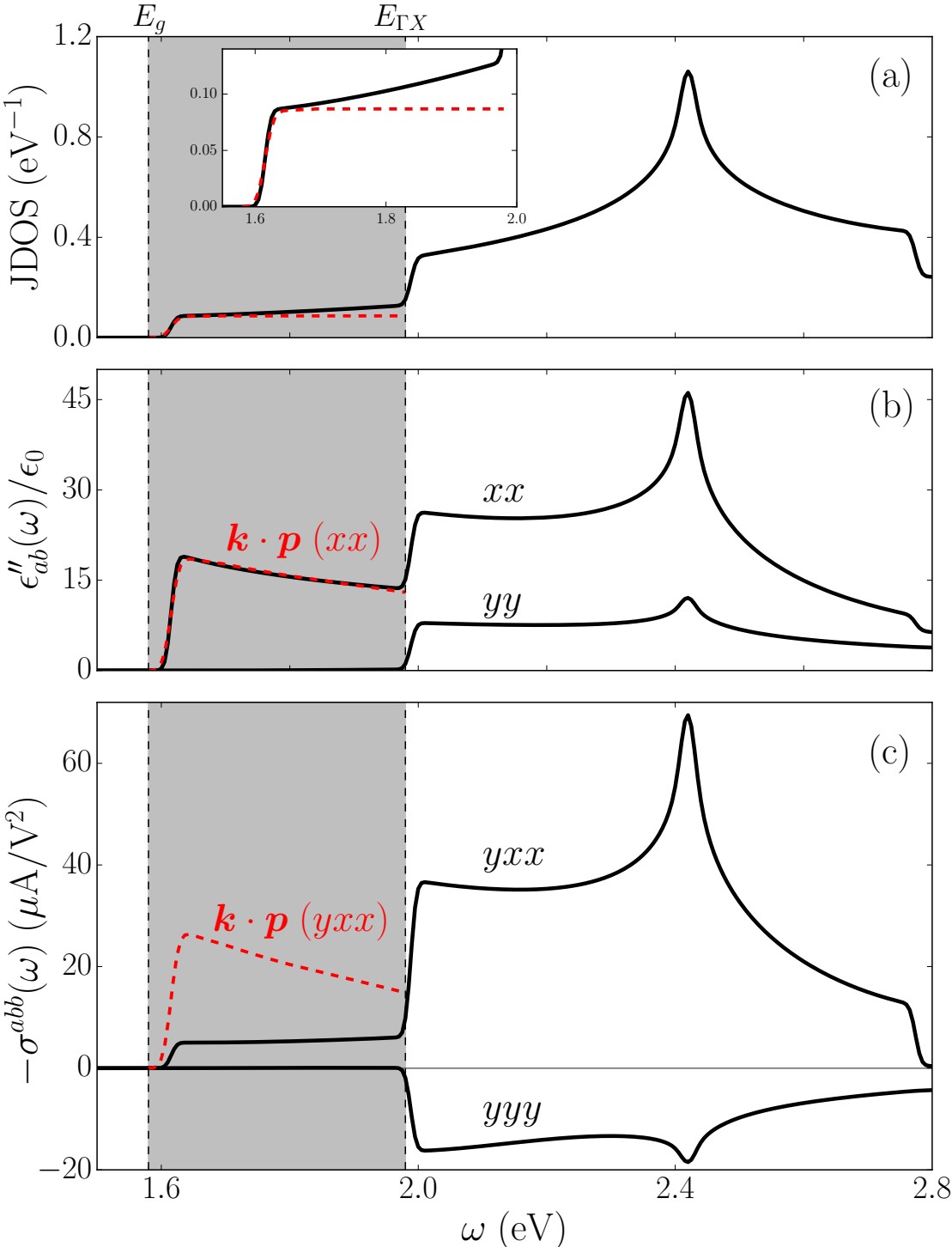

Figure 4: (a) Joint density of states (JDOS), (b) imaginary part of the dielectric function, and (c) shift-current spectrum of monolayer BC$_2$N. Black (solid) lines indicate *ab initio* calculations, while the red (dashed) lines depict the prediction of the two-band $\boldsymbol{k}\cdot\boldsymbol{p}$ model, described in Sec. 4.4. The grey area highlights the band-edge region from $E_g \sim 1.6$ eV to $E_{\Gamma X} \sim 2$ eV, and the inset in (a) shows a blow up of the JDOS in that range.

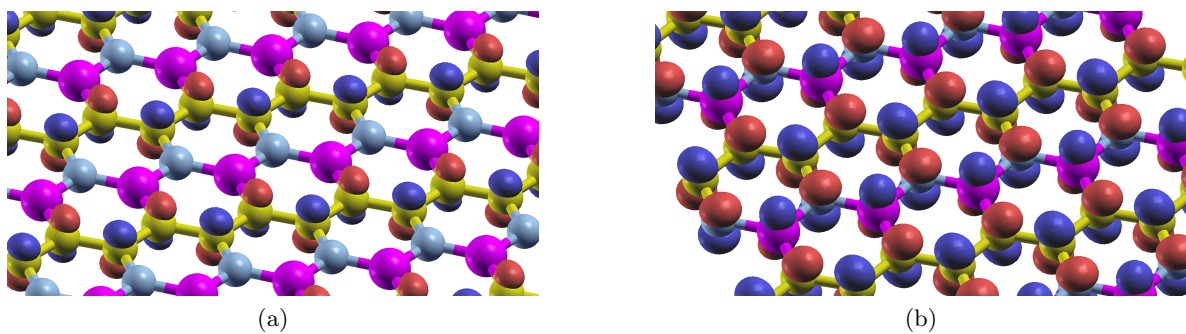

Figure 5: (a) and (b) Isosurfaces of the WFs for the $M = 2$ and $M = 4$ sets, respectively. Red (blue) lobes denote positive (negative) values, and the atoms are color-coded as in Fig. 1.

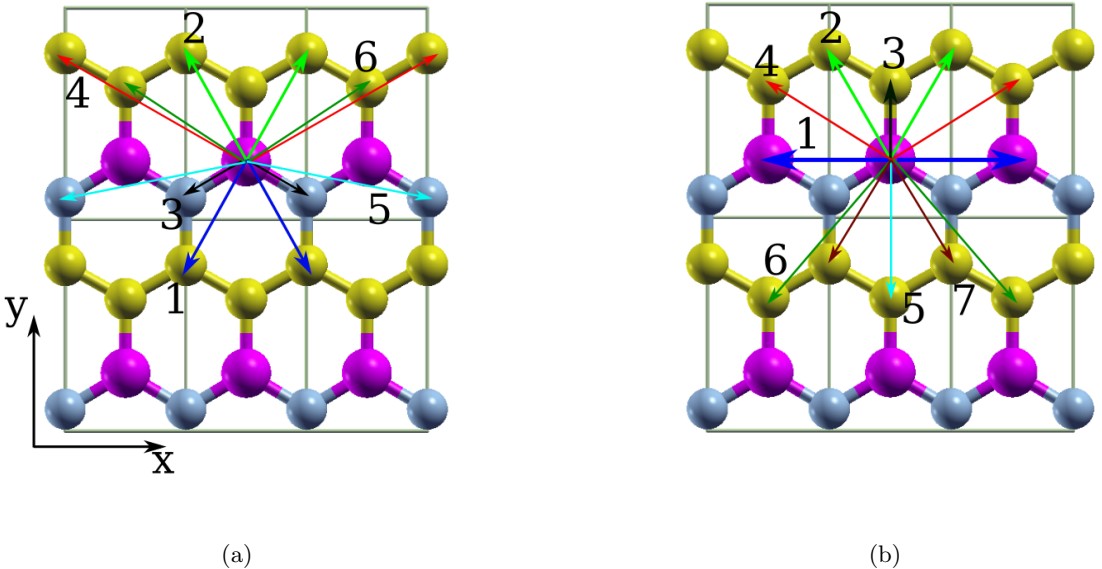

(a)                                                                 (b)

Figure 6: Illustration of largest position matrix elements for the boron atom, labelled in ascending order ($1$ − largest; $2$ − second-largest; $3$ − third-largest; etc.). (a) and (b) show the dominant coefficients for $r^x_{\boldsymbol{R}nm}$ and $r^y_{\boldsymbol{R}nm}$, respectively.

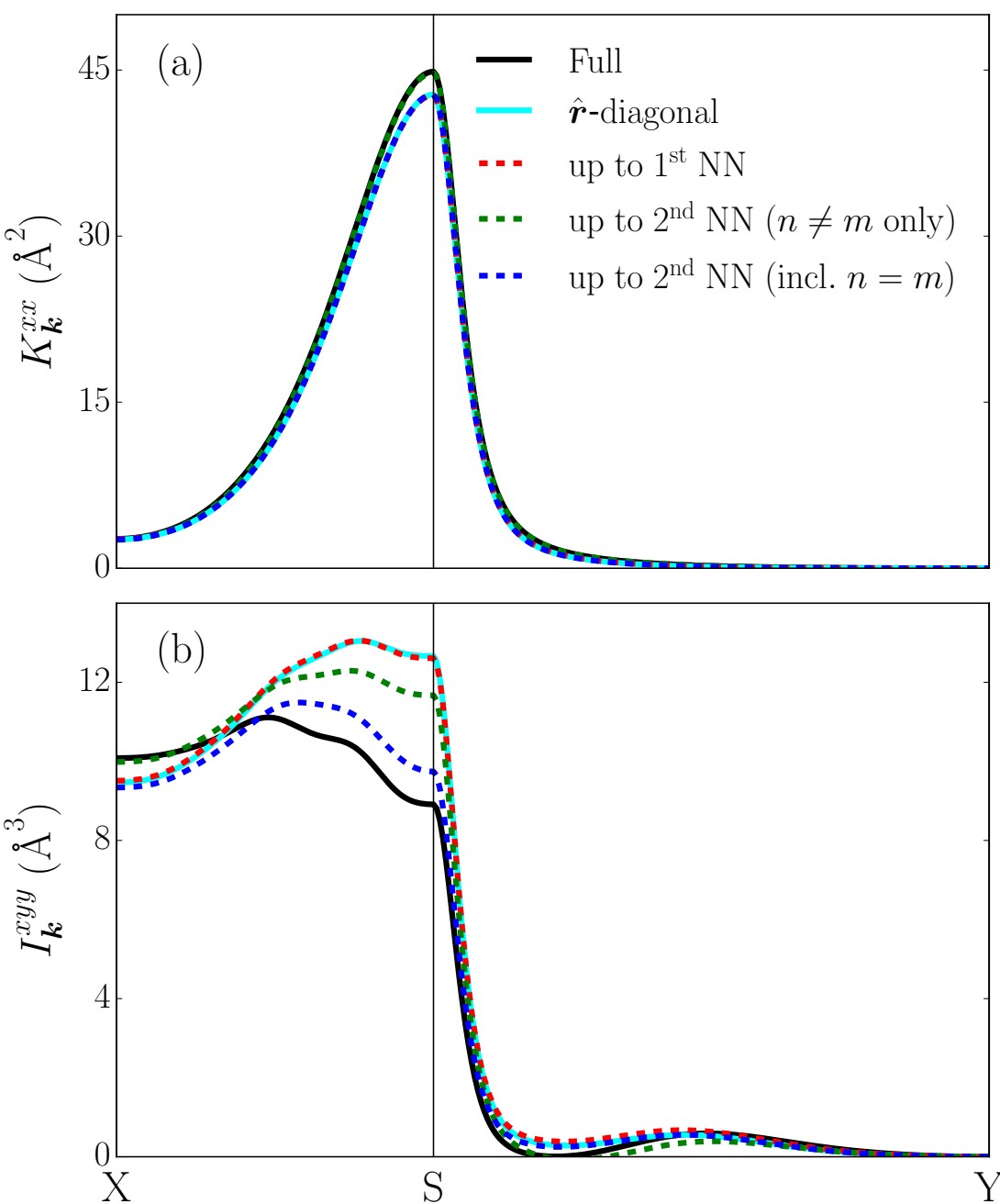

Figure 7: Optical matrix elements calculated along the X–S–Y symmetry lines using a basis with one $p_z$-type WF per atom in the unit cell. (a) and (b) show the linear and quadratic matrix elements of Eqs. 19 and 20, respectively. The different lines correspond to different levels of truncation of the position matrix, keeping up to 2$^{\text{nd}}$ nearest neighbors (see text).

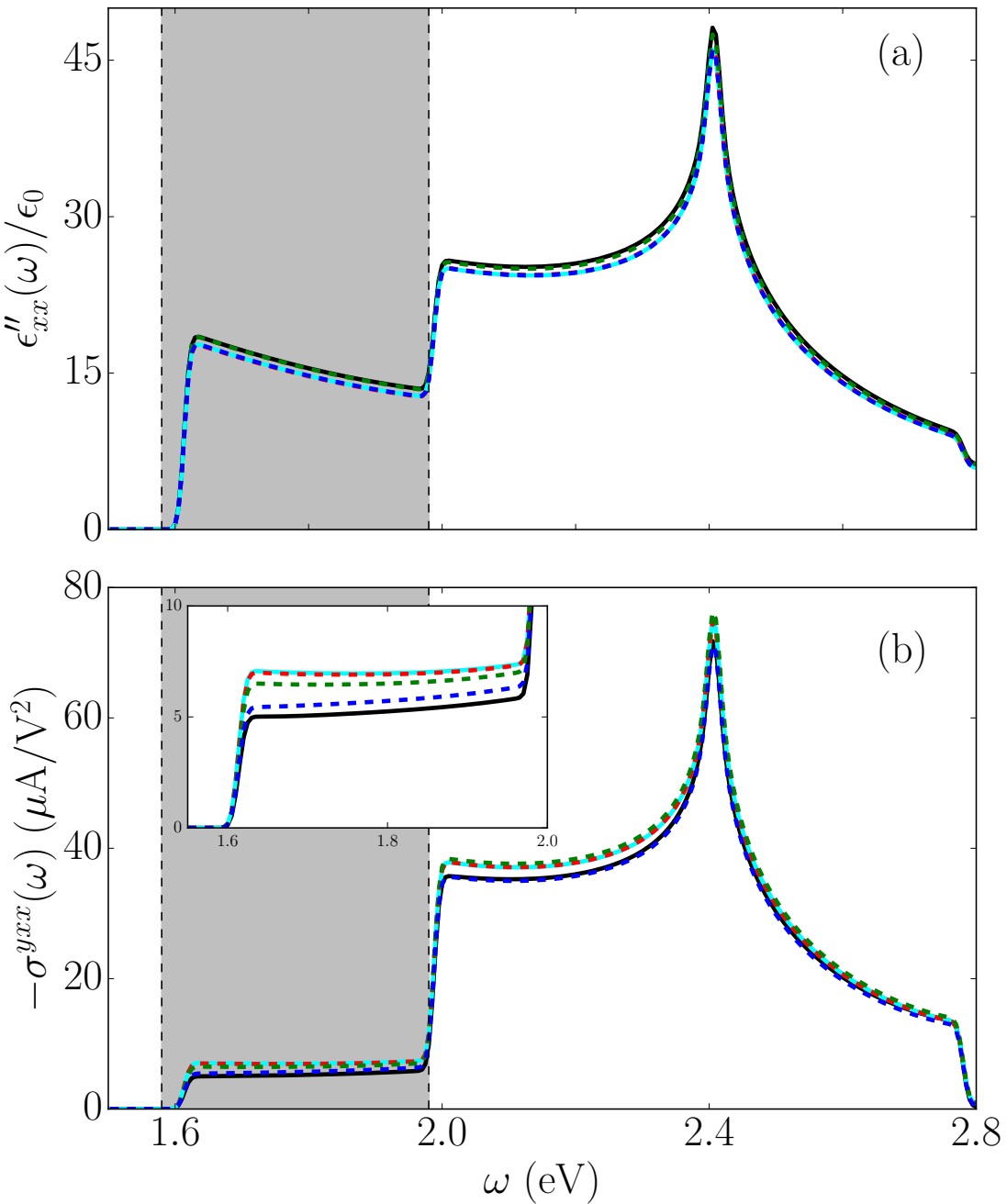

Figure 8: (a) and (b) Spectra of the linear $\epsilon''_{xx}$ and nonlinear $\sigma^{yxx}$ responses, respectively, calculated using a 4-band WF basis. Results obtained considering different levels of truncation of on the position matrix; the labelling scheme is the same as in Fig. 7. The inset in (b) zooms into the band-edge shift photoconductivity.

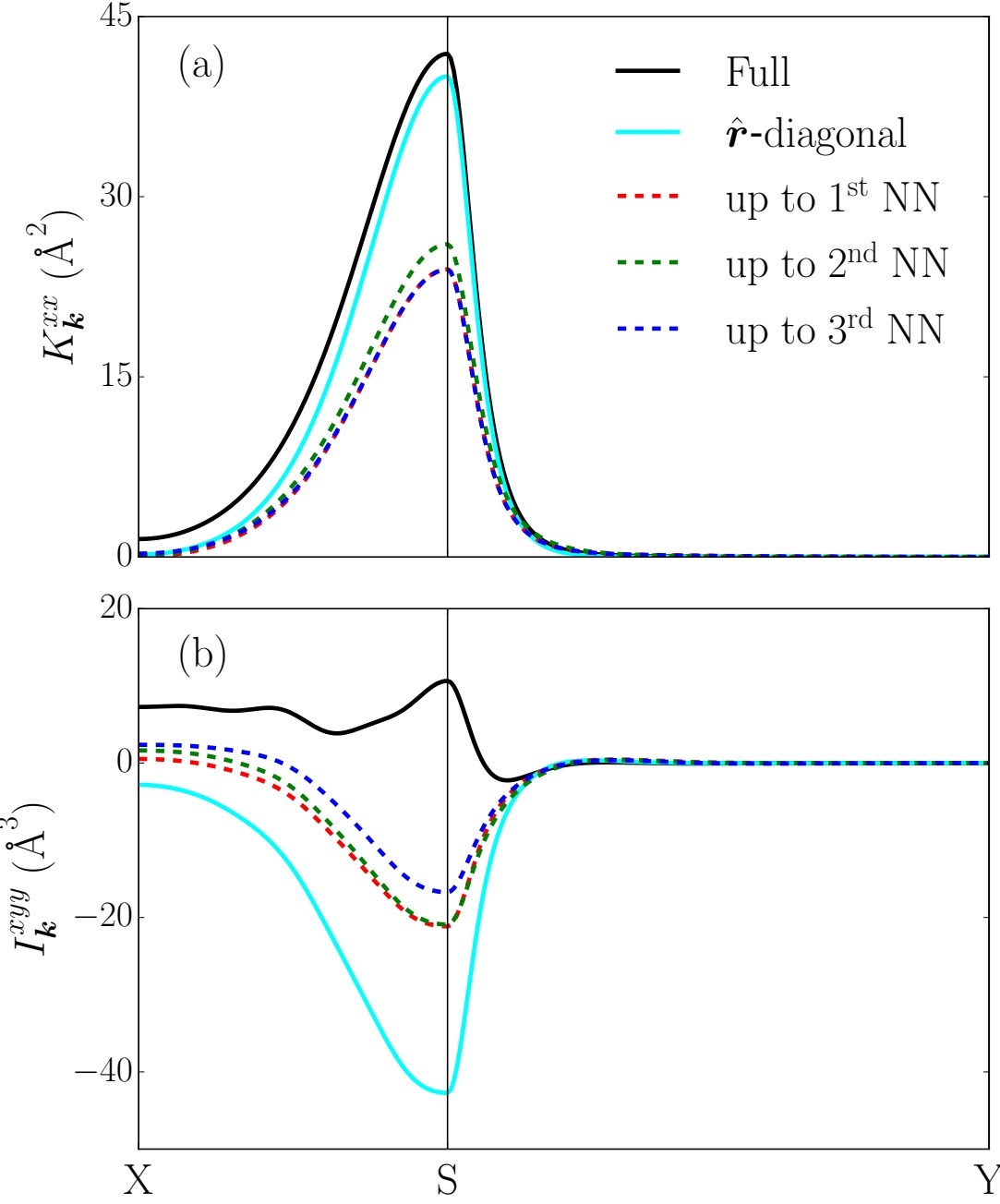

Figure 9: (a) and (b) $k$-resolved plot of transition matrix elements for the linear [Eq. (19)] and nonlinear [Eq. (20)] responses, respectively, calculated using a 2-band WF basis. Results shown for several levels of approximations on the position matrix element up to $3^{\rm rd}$ NN (see text).