# Peer review of "Assessing the role of interatomic position matrix elements in tight-binding calculations of optical properties"

_SciPost Physics_

## Round 2 · Referee Report · Jae-Mo Lihm (Referee 1) · 2021-8-8

Strengths

1 - Reports an important and novel finding on the importance of off-diagonal position matrix elements
2 - The formalism and computational details are presented clearly.

Weaknesses

1 - The relation between the k dot p model and the truncation of position matrix elements is not clear.

Report

This manuscript presents a numerical study on the role the off-diagonal position matrix elements play in the calculation of the linear and nonlinear optical responses of solids. The authors take the case of BC2N to show that the nonlinear shift current response is significantly affected by the truncation of position matrix elements. The authors also demonstrate that the r-diagonal approximation built in the k dot p model leads to inaccuracies in the calculation of shift current.

The finding that the off-diagonal position matrix elements can have substantial quantitative and qualitative effects is a novel discovery that is interesting by itself and should be kept in mind in computational studies on the optical response of solids. Therefore, I would be happy to support the publication of the manuscript if the authors consider the comments below.

  1. The calculation of position matrix elements in the WF basis requires some care. First, did the authors use the finite-difference derivative formula of Ref. [21] to calculate the position matrix elements? If so, there is a finite-difference error. How does it affect the results? Especially, is the hierarchy of the off-diagonal matrix elements robust against this error? Second, the diagonal part of the diagonal position matrix elements can be calculated in two ways (Eq. (22) and Eq. (31) of Ref. [21]). Both are implemented in Wannier90. Which one did the authors use, and does the choice affect the results?

  2. Providing more details on the constructed WFs will be beneficial. First, what are the spreads of the WFs for M=2, 4, and 16? The authors mention the “large spread of the WFs” of the M=2 model (compared to M=4, 16), but the large spread is not evident in Fig. 5. Second, I think it would be useful for each of the WF model to show a comparison between DFT and Wannier bands, together with the corresponding inner and outer windows.

  3. I think that the connection between the k dot p model and the truncation of position matrix elements is not clear. If the k dot p model assumes the r-diagonal approximation, (i) why is the dielectric function very accurate (while Fig. 9(a) shows some error), and (ii) why is the sign of sigma^{yxx} correct, while Fig. 9(b) gives a wrong sign? It would be useful to compare the K and I matrix elements calculated using the k dot p model to those calculated with WFs (with and without the r-diagonal approximation), both analytically and numerically.

  4. The r-diagonal approximation is not an intrinsic limitation of the k dot p model because k dot p model can be constructed directly from ab initio DFT calculation without Wannierization. For example, the interband dipole matrix element at k=0 can be made exact by using the exact velocity matrix elements (dH/dk). Could the authors comment on this point? The authors should also clarify and should provide more details on the statement “We note that this holds true even in the case of k dot p models where the expansion coefficients are extracted from ab initio calculations.”

  5. Would it be possible to improve the WF-derived k dot p model by using the position matrix elements in the construction of the k dot p model?

  6. Which WF model (M=2 or 4 or 16) was used to generate the k dot p model?

  7. Figure 9 shows that matrix elements beyond 3rd NN give a sizable contribution for the M=2 case. How large are the matrix elements themselves? It may be useful to show the decay of the matrix elements (R versus r_R) for M=2, 4, and 16.

  8. In Figure 7, does "incl. n=m" means that only n=m, not n/=m, components of the 2nd NN matrix elements are included? If so, one could write "n=m only", consistently with "n/=m only". If not, then why is the error in the K^{xx} matrix element for "2nd NN (incl. n=m)" greater than "2nd NN (n/=m only)", even though more matrix elements are included? (I would expect that those two cases give the same K^{xx} because r^x is zero for 2nd NN with n=m.)

  9. Do the authors have an explanation why k points far from S are less affected by the truncation?

  10. How do the optical spectra computed using the M=2 model with and without truncation look like? Did the authors consider showing a figure corresponding to Fig. 8 for the M=2 case?

  11. Do the authors have any explanation for the "accidental" agreement between the r-diagonal approximation and the exact value for K in the M=2 case? Is this agreement observed/expected in other materials?

  12. The number of neighbors that can be included is limited by the coarse k grid. For the 15151 grid, up to which NN can be included? Figure 9 shows that adding up to third NN is quite far from convergence. Does the 15151 grid give a converged result?

Requested changes

1 - Page 4: The order of indices m and n in the definition of w_nm is not consistent with that in the definition of f_nm. Is this a typo? 2 - Eq. (14): Do not need a line break 3 - In the computational details, the authors should specify (i) the temperature, and (ii) the regularization parameter for the denominators involving intermediate states (Eq. (38) of Ref. [3]) if relevant. 4 - The last paragraph of page 9: Does "those matrix elements" refer to the onsite inter-orbital position matrix elements? Could the authors clarify how the position matrix elements are related to the PDOS? 5 - Page 9, footnote 2: Is it assumed that f(x) is an even function? 6 - Eqs. (19, 20): The summation notation is not defined. Does it mean that n and m are restricted to the valence and conduction bands, respectively? 7 - Page 10: Does “Here the summations are over the upper valence and lower conduction bands only” mean that n and m in Eqs. (19, 20) are limited to the highest valence band and the lowest conduction band? If so, one could write just write K_k = K_{k,v,c} (with v and c defined as the index of the highest valence band and the lowest conduction band, respectively) without the summation. If not, the authors should clarify the meaning of that statement. 8 - Eq. (44): Do not need a line break. 9 - Figure 5: The figure shows repeated periodic images of the WFs. It would be better to plot only a single WF without its periodic images. If this option is not feasible, the authors should denote a single WF and state that others are periodic images. 10 - Figure 7(a), 8(a): The red dashed line is not visible. 11 - Figure 7(b), 9(b): The y labels seem to be wrong: according to the main text, they should be I^{yxx}.

  • validity: high
  • significance: high
  • originality: top
  • clarity: high
  • formatting: excellent
  • grammar: perfect

Author:  Julen Ibanez  on 2021-12-02  [id 1999]

(in reply to Report 1 by Jae-Mo Lihm on 2021-08-08)
Category:
answer to question

We would like to thank referee Jae-Mo Lihm for his detailed reading of our manuscript and his many relevant comments. Below we reply to each one of them in turn.

Q1. Yes, we used the finite-difference expression of Ref. [21] as implemented in Wannier90 and made sure to use a converged k-mesh. To give a sense of the convergence, the magnitude of position-matrix elements changes by less than 1% when increasing the k-mesh from 15x15x1 to 20x20x1, and even less for finer meshes. This small variation does not affect the hierarchy of the leading position-matrix elements; it may alter the hierarchy of some of the smallest terms, but these are unimportant for the purpose of the manuscript.

As for the diagonal contribution, we calculated it using the default option in the postw90 module of Wannier90, namely Eq. (22), since the other option appears not to be documented. In any case, the values differ by roughly 1% when computed via Eq. (31) for the 15x15x1 k-mesh, and the difference decreases with increasing k-mesh. In general, the influence of the choice on the shift current is on the same order, although we found it affects more the value right at the band edge (roughly 10%). We traced back the reason for this increase to the following: the dominant pieces of the shift current at the band edge turn out to be those that depend on the diagonal parts of the Berry connection (second bit of r.h.s. in Eq. 34 and last two bits in Eq. 36 in Ref. 3 [Ref. 22 in current version]), which are affected roughly 1% by the choice. But since their magnitude is almost one order larger than the total shift-current magnitude (meaning that the sum of the rest of terms, which are not affected by the choice, contribute with opposite sign), that relative percentage increases to roughly 10% when considering the full shift current magnitude. In any case, the importance of the r-diagonal approximation as well as the contribution of the NNs analyzed in the main text is virtually unaffected by the choice. Therefore, and since the use of Eq. (22) appears to be the standard and most tested option, we decided to keep the results.

Q2. In the revised version, we have omitted the 16-WF basis set since we have oriented the focus of the manuscript exclusively on the role of interatomic position matrix elements, for which case the 2- and 4-WF basis sets suffice. We have specified the corresponding spreads of these two sets as well as the comparison of the DFT and Wannier bands in Sec. 4.

Q3. We agree that the scope of the k.p model and connection to the so-called Wannier interpolation scheme was not stated clearly, as also pointed by Referee 2. We have considerably rewritten the manuscript in order to clarify this connection. In particular, see Sec. 5.3.2 of the revised version.

Regarding the numerical results, we have fixed a bug in the way we were extracting the matrix elements of the 2-band case (Fig 9). These are now more in line with those of the k.p model, and, to some extent, also with those of the 4-band case.

Q4. If one uses a complete Bloch basis, then all the information needed to describe the optical response is in the Hamiltonian and its k-space derivatives. In this case, what the Referee mentions holds true and the k.p model does not suffer from the r-diagonal approximation. However, when one uses a Wannier basis set spanning a finite number of bands as in our work, the dipole matrix element acquires a dependence on the position operator. By construction, the k.p model is completely defined by the Hamiltonian and its derivatives, hence it does not capture the piece depending on position.

This should not be interpreted as an argument against the use of localized basis sets; the latter are actually heavily used in the literature, e.g. in the tight-binding approach. But it does point out that one should be careful when describing optical responses using localized basis sets.

Q5. In a sense, this is precisely what the Wannier interpolation method represents; an improved k.p model that incorporates the effect of position.

Q6. M=4. We note that using larger basis sets does not change appreciably the results.

Q7. In the revised version we have included the decay of all position matrix elements versus the orbital distance (see Figs. 5 and 8).

Q8. We thank the referee for spotting this issue. "incl. n=m" means n\neqm plus n=m, so the nomenclature was correct, but there was an error in the computation of the "2nd NN (incl. n=m)" contribution that has been fixed. In the revised version of the manuscript, we have found it more convenient not to separate out 2nd NN terms into n=m and n\=m pieces, and instead we have included 3rd NN terms into the plot.

Q9. We note that the matrix elements themselves are much larger along SX than along SY (specially in the case of the shift current), which partly explains the difference. In addition, inspection of the orbital character of valence and conduction bands reveals that SX is dominated by Carbon atoms, whereas SY has roughly equal contribution from all atoms. The 1st NN position-matrix element along x that joins the two inequivalent Carbon atoms dominates at that level, which might partly explain the larger effect of truncation along SX. However, we find it difficult to find a general and simple reason to explain the behavior of rather complicated transition matrix elements.

Q10. The optical spectra at the band-edge show behavior analogous to the matrix elements at S (Fig. 9). We have considered including a figure analogous to Fig. 8 of the previous version (Fig. 7 in the current version), but we believe it is rather redundant as it shows essentially no new information, hence we have decided not to include it.

Q11. After fixing the bug mentioned in the reply to Q3, the accidental agreement is not present any more.

Q12. In the 2-band case (Fig. 8), it is clearly seen that terms at D~15 Ang. are much smaller than the dominant ones. Given that the Wannier interpolation calculation takes into account terms up to 35 Ang. for the 15*15*1 grid, the contribution of position matrix elements is definitely converged. In any case, we decided to use the 20*20*1 grid to be on the safe side.

Requested changes

RC1. We believe it is not a typo, as this equation agrees with Eq. (57) of the original reference (Ref. [10]).
RC2. Ok.
RC3. We now state in Section 4 that the temperature is set to T=0, and that the regularization parameter is set to 0.04 eV following Ref. [3] (Ref. [22] in the current version).
RC4. Indeed, there is no obvious way of relating the PDOS to position matrix elements, we thank the referee for noting this loose statement. In any case, this sentence referred to the set of 16 WF which has not been included in the revised version, hence this discussion is no longer present.
RC5. Yes, it is assumed. Since the WFs are virtually pz orbitals, these are even in x and y coordinates, hence this assumption is justified. Note however that this footnote is not present in the revised version since the Table to which it referred has been substituted by a plot of the decay of the H and r hoppings.
RC6. Yes, valence and conduction bands are labelled in the text as “v” and “c”, respectively. In any case, the summation does not appear in the revised version.
RC7. Following the suggestion, we have omitted the summation and skipped the use of the K and I symbols, replacing them with the actual transition matrix elements involving the dipole term and (in the case of the shift current) its generalized derivative.
RC8. Ok
RC9. Given that in the revised version we explicitly show the decay of all position matrix elements, we see no more the need of representing the WFs in real space, hence we have omitted that figure.
RC10. Fixed.
RC11. We thank the referee for spotting this typo, we have fixed it.

---

## Round 2 · Referee Report · Anonymous (Referee 2) · 2021-8-16

Strengths

  1. The paper finds a significant disagreement between simplified models and ab initio calculations of shift current in a regime when the simple models are correct for other physical quantities.
  2. The reason for that sensitivity of shift currents to the r-diagonal approximation is explained clearly in terms of the truncated matrix elements.
  3. The paper shows an impressive degree of physical insight into the approximations involved in how the real-space content of orbitals enters into standard tight-binding methods.
  4. Regarding acceptance criterion 2, it has long been known in the community that ab initio calculations reported for second order optical properties are less accurate than for many other quantities. The present work represents a significant step in rectifying that problem. It also represents a useful link between the model and ab initio communities (criterion 3) working on this problem.

Weaknesses

  1. Not a significant weakness, but the paper mostly applies a method developed in a recent publication by two of the authors. The detailed application to shift current is new and important.
  2. It would be nice to have some general guidance on which materials/bonds might make the r-diagonal approximation particularly dangerous, if it is possible to generalize beyond the one example studied in detail here.

Report

This manuscript considers the role of off-diagonal matrix elements of the position operator in determining the optical properties, especially the shift current, of acentric (inversion-breaking) materials. Using ab initio calculations on one material, BC2N, it finds that certain well-known approximations such as the k dot p method are notably less successful for the shift current than for standard properties, and gives a clear explanation of why this is likely to be the case more generally, based on which off-diagonal matrix element is being set to zero for different quantities. In fact, even in many recent ab initio calculations coming out now on specific materials, the important off-diagonal matrix elements are being set to zero, and perhaps this could be called out a little more clearly in the abstract ("a common approximation even in ab initio calculations" or similar), as I fear people carrying out such calculations may not otherwise grasp that this work suggests a possible source of error.

I believe that this is a significant contribution to the literature on an important current topic. Some of the new concepts were already in Reference 3 by an overlapping group of authors, but I feel that moving to shift currents and understanding in detail how simpler approximations fail is a valuable step on its own. In particular, it seems important that even though the k dot p approach works well for other quantities, it does not work well for shift currents at the band edge, but the band edge is where one might wish to drive shift currents for other reasons, such as the large JDOS.

The technical part seemed convincing and well written. I am a little unsure about whether everyone means the same thing by "k dot p model", and in particular whether one could improve the k dot p model with exact matrix elements in order to remove the discrepancy.

One highly cited work on ab initio computation of shift photocurrents is Steve M. Young and Andrew M. Rappe, Phys. Rev. Lett. 109, 116601. If the authors have a comment on whether there is a physical difference in the treatment of position operator matrix elements between that formalism and the present one, I think that would help in uptake of the Wannier interpolation method, assuming it is in fact distinct.

Finally, a comment on a possible future direction: part of the linear dielectric response governs optical activity in materials of low symmetry, and at least some years ago there was a puzzle related to a lack of agreement between ab initio calculations of selenium's optical activity and experimental reality. There is a frank comment in the Erratum to Phys. Rev. Lett. 69, 379 (1992) that some physical ingredient seems to be missing in the ab initio method used in that paper. It is possible that this fact is now understood in the community, but not by me; if it remains an open problem, it would be interesting to see if the method of [3] and this paper resolves the large discrepancy.
  • validity: top
  • significance: high
  • originality: high
  • clarity: top
  • formatting: perfect
  • grammar: excellent

Author:  Julen Ibanez  on 2021-12-02  [id 2000]

(in reply to Report 2 on 2021-08-16)
Category:
answer to question

We thank the referee for his constructive comments on how to improve our manuscript.

We have considerably reworded the abstract, now we explicitly mention the type of calculation where off-diagonal matrix elements are discarded.

Regarding the k.p model, we have explained in detail its scope in the revised version of the manuscript (see text at the beginning of Sec. 5.3.2), as well as in the reply to the Referee 1 (see reply to Q4.).

Regarding the standard reference Phys. Rev. Lett. 109, 116601, it makes use of finite-difference formulas in the ab-initio k-mesh for the calculation of the dipole term and its generalized derivative. Since it does not use a localized basis but works directly with the Bloch eigenstates, the issue of position matrix elements does not arise. The basic difference with respect to our approach is that Wannier interpolation allows computing k-space derivatives of these quantities in a fine k-mesh with low computational cost.

Regarding the comment on selenium's optical activity, it is an interesting puzzle we were unaware of. However, we believe the most important effect causing this discrepancy is not going to be the careful treatment of the position operator, but rather the fact that in PRL 69, 379 (1992) the authors calculate the interband contribution to optical activity only, assuming elemental Se is an insulator, but an intraband contribution due to extrinsic doping can also be present. This problem was addressed for the related compound Te in PRB 97, 035158 (2018), where it was shown (Fig. 12) that the intraband contribution can indeed lead to significant changes to the optical activity. The influence of the different approximations for the position operator in optical activity does remain an interesting open problem for future work.

---

## Round 3 · Referee Report · Jae-Mo Lihm (Referee 1) · 2021-12-6

Report

The authors have responded to my comments adequately. I believe this work on the interatomic position matrix elements will encourage a push to study and include its effect on the calculation of various linear and nonlinear response properties in real materials and thus recommend publication in SciPost Physics.

I have only three minor comments.

1. Regarding the authors' reply on RC1, Ref. [14] (Ref. [10] in the previous version) defines $\omega_{mn} = \omega_m-\omega_n$ (Below Eq. (32), p. 5341), which is different from the authors' definition of $\hbar \omega_{nm} = E_m-E_n$ (below Eq. (4) of the current version). But the form of Eq. (57) of Ref. [14] is identical to Eq. (8) of the submitted manuscript, which means that the delta function reads $\delta(\omega_m - \omega_n - \omega)$ in Ref. [14] and $\delta(\omega_n - \omega_m - \omega)$ in the submitted manuscript. Could the authors check that this is correct?

2. In Section 5.3.2, the coefficients f_i, f_ia, and f_iab are used in Eqs. (20, 21) but not defined in the main text. It will be easier to follow if these coefficients as well as the k dot p Hamiltonian are defined in the main text, rather than in the appendix.

3. Typo in p.16: “sectot” -> “sector”

  • validity: top
  • significance: high
  • originality: top
  • clarity: top
  • formatting: perfect
  • grammar: perfect

Author:  Julen Ibanez  on 2021-12-16  [id 2033]

(in reply to Report 1 by Jae-Mo Lihm on 2021-12-06)

We thank referee Jae-Mo Lihm for his comments, which we have taken into account for the revised version of the manuscript. Here is our reply:

1- Following Appendix A of Ref 22, it can be seen that the order of the nm indexes inside the delta function does not alter the result of the shift current. However, we do agree with the referee that the convention $\omega_{nm}=\omega_{m}-\omega_{n}$, which was also used in Ref. 22, is somewhat anti-intuitive, hence we decided to switch to the more common convention $\omega_{nm}=\omega_{n}-\omega_{m}$.

2- We now define the k.p Hamiltonian and the expansion coefficients in the main text.

3- We have fixed the typo.

---

## Round 3 · Referee Report · Anonymous (Referee 2) · 2021-12-13

Strengths

The response of the authors to my comments and those of the other referee is satisfactory, and I believe that the paper is a meaningful advance in the understanding of the microscopic origin of optical response in materials.

Report

The manuscript meets the standards of SciPost Physics in its current form and can be published as is, in my opinion.

---

## Round 3 · Author Response

We thank the editor in charge for the helpful processing of our manuscript. We also wish to thank both referees for their thoughtful reports. We have modified the manuscript in order to clarify the points raised by the referees, which we believe has resulted in an improved version.

---

## Round 3 · List of Changes

- The Abstract now highlights the importance of interatomic position matrix elements.
- The introduction now comments also on Hamiltonian on-site and hopping matrix elements, and draws analogies between Hamiltonian and position hoppings.
- Section 2 of previous version has been restructured into Sections 2 and 3.
- The main results are now discussed in a different order; Section 5.2 is devoted to the basis set composed of 4 Wannier functions, while Section 5.3 focuses on the set composed of 2 Wannier functions, including discussion of the k.p model.
- The discussion section focuses on hopping terms of the position operator and their relevance.
- Figures and Table: Table 1, Fig. 5 and Fig. 6 of previous version are not present in the modified version, while Fig. 5 and Fig. 8 of the modified version are new. The rest of figures have been slightly modified to address the comments of the referees.

---

## Editorial Decision

resubmitted